# Attention Modulates Electrophysiological Responses to Simultaneous Music and Language Syntax Processing

**DOI:** 10.3390/brainsci9110305

**Published:** 2019-11-01

**Authors:** Daniel J. Lee, Harim Jung, Psyche Loui

**Affiliations:** 1Department of Psychology, Wesleyan University, Middletown, CT 06459, USA; jdlee@wesleyan.edu (D.J.L.); hjung01@wesleyan.edu (H.J.); 2Department of Music, Northeastern University, Boston, MA 02115, USA

**Keywords:** music, language, syntax, attention, comprehension, electroencephalography, event-related potentials

## Abstract

Music and language are hypothesized to engage the same neural resources, particularly at the level of syntax processing. Recent reports suggest that attention modulates the shared processing of music and language, but the time-course of the effects of attention on music and language syntax processing are yet unclear. In this EEG study we vary top-down attention to language and music, while manipulating the syntactic structure of simultaneously presented musical chord progressions and garden-path sentences in a modified rapid serial visual presentation paradigm. The Early Right Anterior Negativity (ERAN) was observed in response to both attended and unattended musical syntax violations. In contrast, an N400 was only observed in response to attended linguistic syntax violations, and a P3/P600 only in response to attended musical syntax violations. Results suggest that early processing of musical syntax, as indexed by the ERAN, is relatively automatic; however, top-down allocation of attention changes the processing of syntax in both music and language at later stages of cognitive processing.

## 1. Introduction

Music and language are both fundamental to human experience. The two domains, while apparently different, rely on several notable similarities: both exhibit syntactic structure, and both rely on sensory, cognitive, and vocal-motor apparatus of the central nervous system. The nature of this relationship between syntactic structure in language and music, and their underlying neural substrates, is a topic of intense interest to the cognitive and brain sciences community.

The Shared Syntactic Integration Resource Hypothesis (SSIRH [1]) is an influential theoretical account of similarities and differences between cognitive processing for music and language. The SSIRH posits that neural resources for music and language overlap at the level of the syntax; in other words, processing of music and language should interact at the syntactic level, but not at other levels such as semantics or acoustic or phonemic structure.

Support for the SSIRH comes from a variety of behavioral and neural studies. Several studies have presented music and language simultaneously with and without syntactic violations, to test for effects of separate and simultaneous syntax violations on behavioral and neural measures [2,3,4,5,6]. Strong support for the SSIRH comes from a self-paced reading paradigm [2], where sentence segments were presented concurrently with musical chord progressions. One subset of the trials contained syntactic violations in language (garden path sentences), and another subset contained syntactic violations in music (out-of-key chords); a third subset contained simultaneous syntactic violations in both domains. Reaction time results showed that during simultaneous violations of music and language, participants were slowest to respond to the double violation than they were to respond to each violation alone. This superadditive effect was not observed in a control experiment which manipulated the timbre of the music and the semantics of the language.

Although these results seem to offer convincing support for SSIRH, Perruchet and Poulin-Charronnat (2013) [7] showed that under semantic garden path manipulations (as opposed to syntactic garden path manipulations), violations of semantics can also yield the same pattern. Based on these results, Perruchet and Poulin-Charronnat (2013) suggested that increased attentional resources, rather than syntax processing per se, could lead to these statistical interactions. 

The idea that attention can influence the pattern of interaction between music and language processing has since received more support. More recent work has argued that the processing resources of syntax for language and music might both rely on domain-general attentional resources, especially when simultaneously processing music and language in a dual-task situation [3,8]. In that regard, classic theories of attention distinguish between the endogenous, voluntary maintenance of a vigilant state, and the exogenous, involuntary orienting to stimulus events [9]. These attentional systems both affect reaction time, but involve different neural resources and unfold differentially over time [10]. Since reaction time during simultaneous linguistic and musical syntax processing may not readily differentiate between overlapping syntax-specific resources and the engagement of attentional resources, we turned to more time-sensitive measures of neural activity during the processing of music and language. This enables direct comparisons between neural responses to musical syntax violations and neural responses to language syntax violations at multiple time windows throughout the temporal cascade of attentional processes that are triggered during music and language processing. By comparing neural markers of syntax violations in the two domains of music and language, and testing for the interaction between attention and violations in each domain, we can clarify the roles that attentional resources might play in the processing of syntax in music and language. 

The Early Right Anterior Negativity (ERAN) and the Early Left Anterior Negativity (ELAN) are reliably elicited event-related potential (ERP) markers of syntax processing in music and language respectively [5,11,12,13,14]. The ERAN is a frontally-generated negative waveform around 200 ms after the onset of musical syntax violations, whereas the ELAN is an analogous frontally-generated negativity after violations in linguistic syntax, such as violations in word category [15] or phrase structure [16]. Musical syntax processing has been localized to the inferior frontal gyrus (IFG) in magnetoencephalographic (MEG) and fMRI studies [17,18,19,20]. Additional results from ERAN of lesioned patients [21] and in children with Specific Language Impairment [22] have also provided evidence for the reliance of musical syntax processing on classic language-related areas such as the inferior frontal gyrus. The ERAN is also posited as an index of predictive processes in the brain, especially in the case of music, due to its reliance on the formation and subsequent violation of predictions that are learned from exposure to musical sound sequences [23]. Impaired ERAN is observed in adults with lesions in the left inferior frontal gyrus (Broca’s area), which provides additional support for the SSIRH. Importantly, the ERAN is also sensitive to top-down task demands, such as attentional resources devoted to the musical stimuli in contrast to a concurrent, non-musical task [24]. When music and speech were simultaneously presented from different locations, irregular chords elicited an ERAN whereas irregular sentences elicited an ELAN; moreover, the ERAN was slightly reduced when irregular sentences were presented, but only when music was ignored, suggesting that the processing of musical syntax is partially automatic [25]. 

While ERAN and ELAN are markers of syntax processing, semantic processing is indicated by the N400, a negative-going centroparietal waveform beginning around 400–500 ms after a semantic anomaly [26,27]. In addition to being sensitive to semantic content of words, the N400 effect reflects the semantic associations between words and the expectancy for them more generally, showing a larger waveform as an incoming word is unexpected or semantically incongruous with the previous context. In response to ambiguities in linguistic syntax that violate the ongoing context, the P600 is another effect that has also been observed [28]. The P600 is a positive waveform centered around the parietal channels and has been observed during garden path sentences, which are syntactically ambiguous sentences when a newly presented word or words require a reinterpretation of the preceding context [29]. Patel et al. (1998) tested the language-specificity of the P600 by presenting chord progressions in music and garden path sentences in language in separate experiments. Their results showed statistically indistinguishable positive waveforms in the P600 range; in addition, they observed an ERAN-like waveform specifically for music [30]

The P600 is similar in topography and latency to the P3, a complex of positive-going event-related potentials elicited from 300 ms and onwards following an unexpected and task-relevant event. The P3 is separable into two components: P3a, a fronto-central ERP, largest around FCz that reflects novelty processing, and P3b, a later parietally-centered ERP largest around Pz that is more sensitive to motivation and task demands [31]. Patients with frontal lobe lesions show altered habituation of the P3, suggesting that the amplitude of the P3 is subject to frontally-mediated processes [32]. The P3a and P3b have both been observed during top-down attention to syntactically incongruous events in music, and these waveforms are sensitive to different levels and genres of expertise [11,33]. Taken together, the literature suggests two main classes of ERPs during unexpectedness in music and language processing: one class within a relatively early time window of approximately 200 ms (ERAN) and another class during the later time window of 500–600 ms (P3, P600, N4). The earlier class of waveforms are thought to be partially automatic, that is, they are elicited even without top-down attention but their amplitude is modulated by attention. The later class of waveforms is highly sensitive to top-down demands including attention.

In this study we compare ERPs elicited by violations in musical and linguistic syntax, while attention was directed separately towards language or music. We used the stimulus materials from Slevc et al’s (2009), but extended this study by adding a musical analog of the language comprehension task. Thus, across two experiments we were able to compare behavioral results during task-manipulated attention to language and music, while independently manipulating syntax in each domain at a finer timescale in order to test for effects in ERPs that are known markers of syntax processing and attention.

## 2. Materials and Methods

### 2.1. Subjects

Thirty-five undergraduate students from Wesleyan University participated in return for course credit. All participants reported normal hearing. Informed consent was obtained from all subjects as approved by the Ethics Board of Psychology at Wesleyan University. Sixteen students (11 males and 5 females, mean age = 19.63, SD = 2.03) were assigned to the Attend-language group: 15/16 participants in this group reported English as their first language, and 9/16 participants reported prior music training (total mean of training in years = 2.23, SD = 3.42). Nineteen students (8 males and 11 females, mean age = 19.40, SD = 2.03) were assigned to the Attend-music group. Background survey and baseline tests of one participant in this group was missing as the result of a technical error. Of the remaining 18 participants, 12/18 reported English as their first language, and 11/18 participants reported having prior music training (total mean of training in years = 3.11, SD = 4.01). 

The two groups of subjects did not differ in terms of general intellectual ability, as measured by the Shipley Institute of Living scale for measuring intellectual impairment and deterioration [34]. Nor did they differ in low-level pitch discrimination abilities as assessed by a pitch-discrimination task [35]. (Two-up-one-down staircase procedure around the center frequency of 500 Hz showed similar thresholds between the two groups.) They also did not differ in musical ability as assessed using the Montreal Battery for Evaluation of Amusia [36], or in duration of musical training (years of musical training was not different between the two groups, X^2^ = 0.0215, *p* = 0.88). Table 1 shows the demographics and baseline test performance of the participants in both conditions. 

### 2.2. Stimuli

The stimuli were adapted from Slevc, Rosenberg [2]. There were 144 trials in the study, including 48 congruent trials, 48 musically-incongruent trials, and 48 language-incongruent trials. In each trial, an English sentence was presented in segments simultaneously with a musical chord progression. Each segment of a sentence was paired with a chord that followed the rules of Western tonal harmony in the key of C major, played in a grand piano timbre. Linguistic syntax expectancy was manipulated through syntactic garden-path sentences, whereas musical syntax expectancy was manipulated through chords that were either in-key or out-of-key at the highlighted critical region (Figure 1). The chords and sentence segments were presented at the regular inter-onset interval of 1200 ms. At the end of each sentence and chord progression, a yes/no comprehension question was presented on the screen: In the Attend-Language group, this question was about the content of the sentence to direct participants’ attention to language (e.g., “Did the attorney think that the defendant was guilty?”). For the Attend-Music group, the question at the end of the trial asked about the content of the music (e.g., “Did the music sound good?”) to direct participants’ attention to the music. Participants were randomly assigned to Attend-Language and Attend-Music groups. Participants’ task, in both Attend-Language and Attend-Music groups, was always to respond to the question at the end of each trial by choosing “Yes” or “No” on the screen. 

### 2.3. Procedure

Participants first gave informed consent and filled out a background survey on their musical training, as well as a battery of behavioral tasks including the Shipley Institute of Living Scale to screen for impairments in intellectual functioning [34], the Montreal Battery for Evaluation of Amusia (MBEA) to screen for impairments in musical functioning [36], and a pitch discrimination test as a three-up-one-down psychophysical staircase procedure around the center frequency of 500 Hz to assess pitch discrimination accuracy [35]. The experiment was run on a Macbook Pro laptop computer using Max/MSP [37]. At the start of the experiment, participants were told to pay attention to every trial, and to answer a yes-or-no comprehension question about the language (Attend-Language condition) or about the music (Attend-Music condition) at the end of each trial. They were given a short practice run of 5 trials of the experiment in order to familiarize themselves with the task before EEG recording began. EEG was recorded using PyCorder software from a 64-channel BrainVision actiCHamp setup with electrodes corresponding to the international 10–20 EEG system. Impedance was kept below 10 kOhms. The recording was continuous with a raw sampling rate of 1000 Hz. EEG recording took place in a sound attenuated, electrically shielded chamber. 

### 2.4. Behavioral Data Analysis

Behavioral data from Max/MSP were imported to Excel to compute the accuracy of each participant. Accuracy was evaluated against 50% chance-level in one-sample two-tailed t-tests in SPSS. For the Attend-Music condition, two subjects’ behavioral data were lost due to technical error.

### 2.5. EEG Preprocessing

BrainVision Analyzer software (Brain Product Gmbh) 2.1 was used to preprocess raw data. EEG data were first re-referenced to TP9 and TP10 mastoid electrodes, and filtered with high-pass cutoff of 0.5 Hz, low-pass cutoff of 30 Hz, roll-off of 24 dB/oct, and a notch filter of 60 Hz. These filter settings were chosen based on previous work that looked at target ERPs similar to the current study [33,38], since filter settings introduce artifacts in ERP data [39]. Ocular correction ICA was applied to remove eye artifacts for each subject. Raw data inspection was done semi-automatically by first setting maximal allowed voltage step as 200 µV/ms, maximal difference of values over a 200 ms interval as 400 µV, and maximal absolute amplitude as 400 µV. Then, manual data inspection was performed to remove segments with noise due to physical movements. 

### 2.6. Event-Related Potential Analysis

The preprocessed data were segmented into four conditions: music congruent, music incongruent, language congruent, and language incongruent. Each segment was 1200 ms long, spanning from a 200 ms baseline before the onset of the stimulus to 1000 ms after stimulus onset. The segments were averaged across trials, baseline-corrected, and grand-averaged across the subjects. To identify effects specific to syntax violations in each modality, a difference wave was created for each violation condition by subtracting ERPs for congruent conditions from ERPs for incongruent conditions, resulting in a Music-specific difference wave (Music violation minus no violation) and a Language-specific difference wave (Language violation minus no violation). From these difference waves we isolated ERP amplitudes at two recording sites, one at each time window of interest: E(L/R)AN from site FCz at 180–280 ms, and the N4 and P3 at site Pz at 500–600 ms. The mean amplitude of each ERP was exported for each participant from BrainVision Analyzer into SPSS for analysis. 

Because both groups of participants experienced both types of syntactic violations (music and language), but each group of participants attended to only one modality (music or language), we used a mixed-effects analysis of variance (ANOVA) with the within-subjects factor of Violation (two levels: music and language) and the between-subjects factor of Attention (two levels: attend-music and attend-language). This was separately tested for the two time windows: 1) the early ERAN/ELAN time window of 180–280 ms, and 2) the later N4/P3 time window of 500–600 ms. 

## 3. Results

### 3.1. Behavioral Results

Participants performed well above the 50% chance level on language comprehension questions during the Attend-Language condition (M = 0.8457, SD = 0.0703, two-tailed t-test against chance level of 50% correct: *t*(15) = 19.661, *p* < 0.001), and on music comprehension questions during the Attend-Music condition (M = 0.6631, SD = 0.1253, two-tailed t-test against chance level of 50% correct: *t*(16) = 5.371, *p* < 0.001). This confirms that participants successfully attended to both language and music stimuli. Participants performed better on the Attend-Language than on the Attend-Music questions (*t*(31) = 5.12, *p* < 0.001).

### 3.2. Event-Related Potentials

Figure 2 shows each ERP and scalp topographies of difference waves. Figure 3 shows the specific effects for each ERP; statistics are shown in Table 2. A right anterior negative waveform was observed 180–210 ms following music violations, consistent with the ERAN. This ERAN was observed during both Attend-Language and Attend-Music conditions. During the Attend-Language condition, a centroparietal negative waveform was observed 500–600 ms following language violations, consistent with an N400 effect. This N400 effect was not observed during the Attend-Music condition. Instead, a posterior positive waveform was observed 500–600 ms after music violations during the Attend-Music condition, consistent with the Late Positive Complex or the P3 or P600 effect. These selective effects are tested in a mixed-model ANOVA for each time-point, with a between-subjects factor of attention (two levels: attend-music vs. attend-language) and a within-subjects factor of modality of syntax violation (two levels: music and language), as described below. 

*180–280 ms:* A significant negative waveform was observed for the music violation but not for the language violation. The within-subjects effect of Violation showed a significant difference between music and language violations (F(1,33) = 33.198, *p* < 0.001). The between-subjects effect of Attention was not significant (F(1,33) = 1.381, *p* = 0.248). There was no significant interaction between the Violation and Attention factors (Figure 2). Tests of between-subjects effects showed no significant difference between the Attend-Music and the Attend-Language conditions (Figure 3). 

*500–600 ms.* For the late time window, the within-subjects effect of Violation was significant (F(1,33) = 31.317, *p* < 0.001), and the between-subjects effect of Attention was significant (F(1,33) = 9.763, *p* = 0.004). Here, an Attention by Violation interaction was also significant (F(1,33) = 9.951, *p* = 0.003). This interaction is visible in the ERP traces and topographic plots in Figure 2 as well as in the amplitude results plotted in Figure 3: in the Attend-Language condition, only language violations elicited a negative waveform resembling an N400, whereas music violations were no different from the no-violation condition. The N400 shows a latency of 400–800 ms and a centro-parietal topography (Figure 2A), consistent with classic reports of the N400 effect (Kutas and Hillyard, 1984). In contrast, during the Attend-Music condition, only music violations elicited a large positive P3 waveform, whereas language violations showed no difference from the no-violation condition. The P3 shows a latency of 400–1000 ms and a centro-parietal topography (Figure 2B), consistent with the P3b subcomponent of the P3 complex (Polich, 2007). The P3 was only observed for music violations when attention was directed to music, and the N400 was only observed for language violations when attention was directed to language. This attention-dependent double dissociation between P3 and N400 is visible in Figure 2 and Figure 3. 

While musical violations elicited an ERAN in the early time window and a P3 in the late time window, language violations only showed an N400 in the late time window, and no effect in the early time window. One potential explanation is that a minority of participants were not first-language English speakers; these participants may have been less sensitive to syntax violations as manipulated by the garden path sentences. Removing the subjects whose first language was not English resulted in a smaller sample size of all first-language English speakers: *n* = 15 in the Attend-Language condition, and *n* = 12 in the Attend-Music condition. Repeating the above behavioral and ERP analyses on these smaller samples showed the same pattern of results. Behavioral results showed significantly above-chance performance on both Attend-language condition (*t*(14) = 28.66, *p* < 0.001) and Attend-music conditions (*t*(11) = 4.93, *p* = 0.002). ERP statistics for first-language English speakers are shown in Table 3.

## 4. Discussion

By separately manipulating linguistic syntax, musical syntax, and attention via task demands during simultaneous music and language processing, we were able to disentangle the effects of top-down attention on bottom-up processing of syntax and syntactic violations. Three main findings come from the current results: 1) For both music and language, syntactic violation processing activates a cascade of neural events, indexed by early and late ERP components as seen using time-sensitive methods. This replicates prior work (Koelsch et al., 2000 [11,12], Hahne and Friederici, 1999 [13,14], and many others). 2) Early components are less sensitive to attentional manipulation than late components, also replicating prior work [40,41]. 3) Attention affects musical and linguistic syntax processing differently at late time windows. This finding is novel as it extends previous work that identify early and late components in music and language syntax processing, by showing that the late components are most affected by attention, whereas the earlier stages of processing are less so. Taken together, results expand on the SSIRH by showing that top-down manipulations of attention differently affect the bottom-up processing of music and language, with effects of attention becoming more prominent throughout the temporal cascade of neural events that is engaged during music and language processing. We posit that the early stages of processing includes mismatch detection between the perceived and the expected events, with the expectation being core to syntactical knowledge in both language and music. In contrast, the late attention-dependent processes may include cognitive reanalysis, integration, and/or updating processes, which may require general attentional resources but are not specific to linguistic or musical syntax. 

In some respects, the present results add to a modern revision of the classic debate on early- vs. late-selection theories of attention. While early-selection theories (Broadbent, 1958) posited that attention functions as a perceptual filter to select for task-relevant features in the stimulus stream, late-selection theories have provided evidence for relatively intact feature processing until semantic processing [42,43] or until feature integration [44]. Due to their fine temporal resolution, ERP studies provide an ideal window into this debate, allowing researchers to quantify the temporal cascade of neural events that subserve perceptual-cognitive events such as pitch and phoneme perception, and syntax and semantics processing. ERP results from dual-task paradigms such as dichotic listening have shown that attention modulates a broad array of neural processes from early sensory events [45,46] to late cognitive events [47,48]. Here we observe the ERAN in response to musical syntax violations regardless of whether attention was directed to language or to music. The ERAN was elicited for music violations even when in the attend-language condition; furthermore its amplitude was not significantly larger during the attend-music condition. This result differs from previous work showing that the ERAN is larger during attended than during unattended conditions [24]. The difference likely stems from the fact that while in the previous study the visual task and the musical task were temporally uncorrelated, in the present study the language stimuli (sentence segments) and musical stimuli (chords) were simultaneously presented, with each language-music pair appearing in a time-locked fashion. Thus, when in the attend-language condition, the onset of musical chords became predictably coupled with the onset of task-relevant stimuli (sentence segments), even though the musical chords themselves were not task-relevant. This predictable coupling of task-irrelevant musical onsets with task-relevant linguistic stimulus onsets meant that it became more advantageous for subjects to allocate some bottom-up attentional resources to the music, or to allocate attentional resources to all incoming sensory stimuli at precisely those moments in time when stimuli were expected [49], as one modality could help predict the other. The fact that the ERAN was observed even when only slightly attended provides some support for a partially automatic processing of musical syntax, as posited in previous work [24]. When musical syntax violations were not task-relevant but were temporally correlated with task-relevant stimuli, they elicited intact early anterior negativity but no late differences from no-violation conditions. This early-intact and late-attenuated pattern of ERP results is also consistent with the relative attenuation model of attention, which posits that unselected stimulus features are processed with decreasing intensity [50].

One remaining question concerns whether the ERAN is driven by music-syntax violations, or whether the effects may be due to sensory violations alone. Indeed, musical syntax violations often co-occur with low-level sensory violations, such as changes in roughness or sensory dissonance. In that regard, the musical syntax violations used in the present study are carefully constructed to avoid sensory dissonance and roughness (see supplementary materials of Slevc et al., 2009 for a list of chord stimuli used). Thus the effects cannot be explained by sensory violations. Furthermore, Koelsch et al. (2007) had shown that ERAN is elicited even when irregular chords are not detectable based on sensory violations, which supports the role of ERAN in music-syntax violations. Given our stimuli as well as previous evidence, we believe that the currently observed ERAN reflects music-syntax violations rather than sensory violations. In contrast, no ELAN was observed in response to language violations. This may be because we used garden path stimuli for language violations, while previous studies that elicited early negative-going ERPs used word category violations [14] and phrase structure violations [16] rather than garden path sentences. The introduction of the linguistic garden path requires that participants re-parse the syntactic tree structure during the critical region of the trial; this effort to re-parse the tree likely elicited the N4 at the later time window, but lacks the more perceptual aspect of the violation that likely elicited the ELAN in prior studies (Hahne et al., 1999). Thus, the garden-path sentences and music-syntactic violations used in the present study may have tapped into distinct sub-processes of syntax processing.

It is remarkable that linguistic syntax violations only elicited a significant N400 effect, and no significant effects over any other time windows, even when language was attended. In contrast, musical syntax violations elicited the ERAN as well as the P3 in the attended condition, with the ERAN being observed even when musical syntax was unattended. Note that the P3 effect in this experiment is similar in topography and latency to the P600, which has been observed for semantic processing during garden path sentences. It could also be the Central-Parietal Positivity (CPP), which reflects accumulating evidence for perceptual decisions [51], which can resemble the P3 [52]. During the attend-music condition, linguistic syntax violations elicited no significant ERP components compared to no-violation conditions. This suggests a strong effect of attention on language processing. It is also worth noting that we saw a clear N400 and not a P600 or a P3 in response to garden path sentences in language. The relationship between experimental conditions and N400 vs. P600 or P3 is an ongoing debate in neurolinguistics: Kuperberg (2007) posits that the N400 reflects semantic memory-based mechanisms whereas the P600 reflects prolonged processing of the combinatorial mechanism involved in resolving ambiguities [28]. Others argue that whether an N400 or a P600 is observed may in fact depend on the same latent component structure; in other words, the presence and absence of N400 and P600 may reflect two sides of the same cognitive continuum, rather than two different processes per se [53,54,55]. If the N400 and P600 are indeed two sides of the same coin, then this could mean that language and music processing are also more related than the different effects would otherwise suggest. 

## 5. Limitations 

One caveat is that, similar to the original paradigm from which we borrow in this study [2], music was always presented auditorily, whereas language was always presented visually. Thus, the differences we observe between musical and linguistic syntax violation processing could also be due to differences in the modality of presentation. In future studies it may be possible to reverse the modality of presentation, such as by visually presenting musical notation or images of hand positions on a piano [56] with spoken sentence segments. Although doing so would require a more musically trained subject pool who can read musical notation or understand the images of hand positions, prior ERP studies suggest that visually presented musical-syntactic anomalies would still elicit ERP effects of musical syntax violation, albeit with different topography and latency [56]. Furthermore, although participants performed above chance on both attend-language and attend-music comprehension questions, they did perform better on the attend-language task; this imposes a behavioral confound that may affect these results. Future testing on expert musicians may address this behavioral confound. Future studies may also work to increase the sample size, and to validate and match the samples with sensitive baseline measures in both behavioral and EEG testing in order to minimize confounding factors arising from potential differences between participant groups. Importantly, garden path sentences are only one type of syntactic violation; it remains to be seen how other types of violations in linguistic syntax, such as word category violations, may affect the results. Finally, it is yet unclear how syntax and semantics could be independently manipulated in music, or indeed the degree to which syntax and semantics are fully independent, in music as well as in language [57]. In fact, changing musical syntax most likely affects the meaning participants derive from the music; however, specifically composed pieces with target words in mind might be a way to get at a musical semantics task without overtly manipulating syntax [58]. Nevertheless, by separately manipulating music and language during their simultaneous processing, and crossing these manipulations experimentally with top-down manipulations of attention via task demands, we observe a progressive influence of attention on the temporal cascade of neural events for the processing of music and language.

## Figures and Tables

**Figure 1 brainsci-09-00305-f001:**
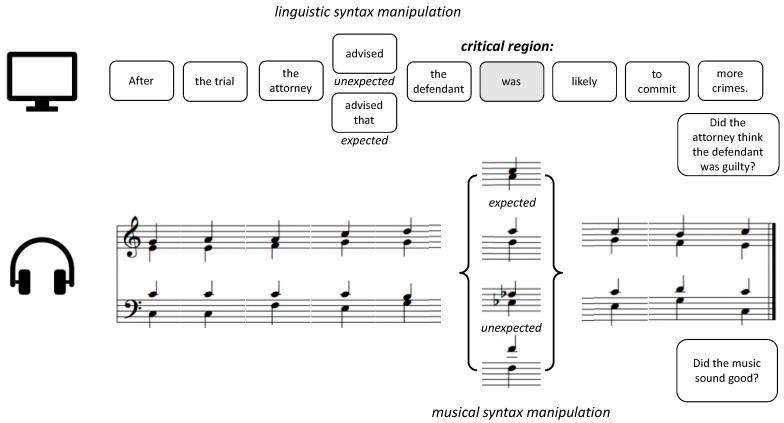
Example trials for Attend-language and Attend-music conditions.

**Figure 2 brainsci-09-00305-f002:**
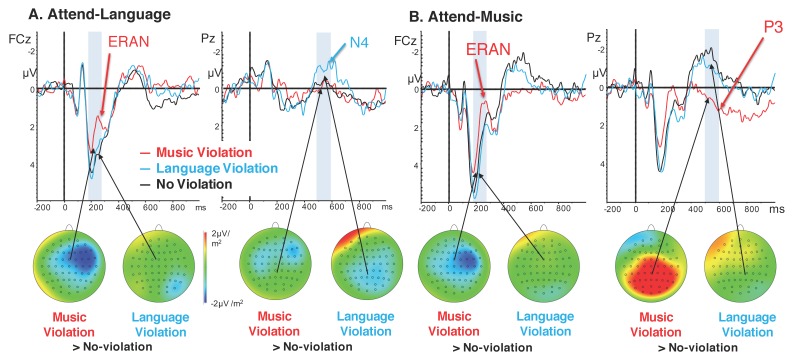
Overlays of ERPs from each condition with topographic maps of the difference wave between violation and no-violation conditions. Music syntax violation condition is shown in red and linguistic syntax violation condition is shown in blue. Black represents a condition when neither stimulus was violated. Topographic plots show difference waves between music violation and no-violation, or between language violation and no-violation. (**A**) When attending to language. (**B)** When attending to music.

**Figure 3 brainsci-09-00305-f003:**
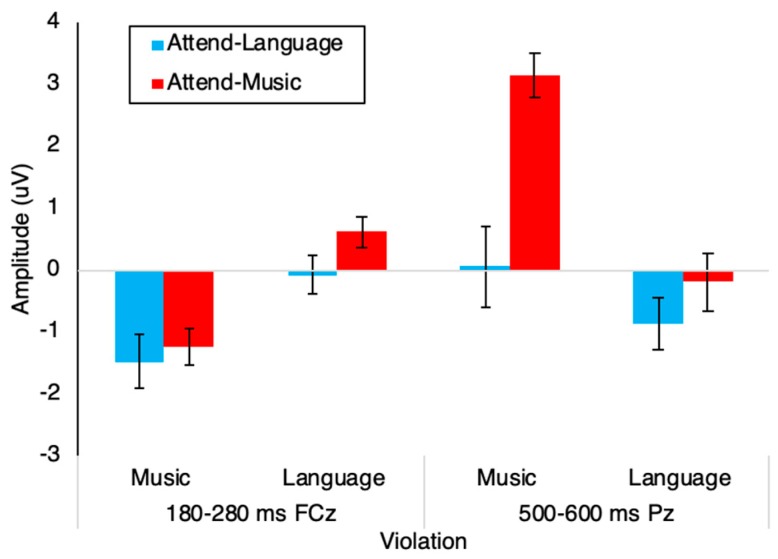
ERP effects of violation (amplitude of difference waves) across different conditions.

**Table 1 brainsci-09-00305-t001:** Demographics and baseline test performance of the participants. Data are shown as mean (SD), range, or proportion. SD: Standard Deviation. *n*: Count in proportion.

Variable	Attend-Language (*N* = 16)	Attend-Music (*N* = 19)
Age in years, M (SD)	19.625 (2.029)	19.389 (2.033)
Male, *n*	11/16	8/19
Music Training, years, M (SD)	2.233 (3.422)	3.105 (4.012)
Musically trained, *n*	9/16	11/18
Full-Scale IQ (Estimated from Shipley-Hartford IQ scale, M (SD))	100 (10)	101 (7)
MBEA, M(SD)	23.375 (3.828)	25.11 (2.685)
Pitch Discrimination, ΔHz/500 Hz, M (SD)	12.469 (11.895)	11.087 (7.174)
Normal Hearing, %	100%	100%
English as First Language, *n*	15/16	12/18

**Table 2 brainsci-09-00305-t002:** ERP statistics.

**Tests of Within-Subjects Contrasts**					
**Source**	**Time-Window**	**df**	**F**	***p***	**Partial eta^2^**
Violation	180–280 ms	1	33.198	< 0.001	0.501
	500–600 ms	1	31.317	< 0.001	0.487
Violation * Attend	180–280 ms	1	0.64	0.43	0.019
	500–600 ms	1	9.951	0.003	0.232
**Tests of Between-Subjects Effects**					
**Source**	**Time-Window**	**df**	**F**	**p**	**Partial eta^2^**
Attend	180–280 ms	1	1.381	0.248	0.040
	500–600 ms	1	9.763	0.004	0.228

**Table 3 brainsci-09-00305-t003:** ERP statistics for first-language English speakers only.

**First Language English Speakers Only**		**Tests of Within-Subjects Contrasts**			
**Source**	**Time-Window**	**df**	**F**	***p***	**Partial eta^2^**
Violation	180–280 ms	1	14.216	< 0.001	0.428
	500–600 ms	1	30.722	< 0.001	0.618
Violation * Attend	180–280 ms	1	0.195	0.664	0.01
	500–600 ms	1	11.075	0.004	0.368
		**Tests of Between-Subjects Effects**			
**Source**	**Time-Window**	**df**	**F**	***p***	**Partial eta^2^**
Attend	180–280 ms	1	0.971	0.337	0.049
	500–600 ms	1	13.99	< 0.001	0.424

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
