# Peer review of "Attention Modulates Electrophysiological Responses to Simultaneous Music and Language Syntax Processing"

_brainsci, 2019, doi:10.3390/brainsci9110305_

Round 1
Reviewer 1 Report
The goal of the study has become much clearer. Particularly the clarification that there was no double violation was important for understanding the rationale and results. While I think that the manuscript has improved, I’d encourage the authors to still sharpen one point in the discussion: After having concluded that attention influences syntactic processing in late rather than early processing stages it would be interesting to get back to the SSIRH that was mentioned in the introduction and comment on how the present findings may relate back to findings of music-syntax interactions. Are interactions at late processing stages (and possibly also behavioural interactions) an “artefact” of general attentional resources? Shall we trust more in interaction effects at early processing stages? I particularly have Slevc & Okada’s recent proposals in mind. What implications (if any) do these data have for that debate? A few comments in these directions would help to balance intro and discussion.
Minor:
Line 88: correct “observed in and in adults” Line 99: correct “and the expectancy for more generally” Line 126: Perhaps rephrase “we adapted Slevc et al’s (2009) paradigm” – this current wording triggers in the reader that you were interested in music-language-syntax interactions. Line 220: Please specify whether the analyses in early and late time windows were done for both Fz and Pz, or on Fz for ERAN and Pz for P3/P600. Line 261: Please change “topo plots” to “topographic plots”. Table 1 and 2: the “eta” sign is missing in partial eta squared. Line 301/302: Please let me explain better – I meant that different time windows reflect different sub-processes of syntax processing: early processes are more related to detection of a mismatch between expected and perceived syntactic category (e.g., word category or chord function), and late processes are more related to reanalysis, integration or repair processes. These latter processes may require more attention than early detection, and may nicely link to your late feature integration discussion in the next paragraph. Maybe simply rephrase this sentence. Line 347: studies that elicited ELAN used “word category violations” (not tense violations). You could also cite Neville et al. (1991, J Cogn Neurosci). Line 384: to address the behavioural confound, it may be worth testing musicians.Author Response
We are pleased that this reviewer found our manuscript to be improved. We agree that we could tie the discussion back into the introduction more. Based on these results, we believe that interactions at late processing stages are likely not restricted to musical or linguistic syntax processing, but may instead belong to more domain-general attention resources. In the present revision we expand on the Discussion section to tie back to the SSIRH and its ensuing debate on attention, before leading in to the implications of the present results for early vs. late selection theories of attention.
Minor:
Line 88: Done.
Line 99: Done.
Line 126: This is now rephrased to show that we are using the stimulus materials from Slevc et al (2009), and then extending the Slevc et al study, rather than simply adapting their paradigm.
Line 220: The analyses were done in only FCz for ERAN (early time window) and only Pz for P3/P600 (late time window). This is now clarified in the Methods section.
Line 261: Done.
Table 1 and 2: We see the eta sign on our copy. Perhaps it depends on whether the Symbol font is installed on each computer? We now change the eta sign to be written out in the same font as the rest of the tables.
Line 301/302: We thank the reviewer for this insightful suggestion. We agree with the reviewer’s characterization of the early and late processes as mismatch detection and reanalysis/integration, and we have now integrated this point into the Discussion section.
Line 347: We thank the reviewer for helping us clarify this. We have now changed our description of ELAN findings to word category rather than tense violations. Thank you also for pointing us to Neville et al (1991). We now cite this highly relevant work in the Introduction and Discussion.
Line 384: We agree with the reviewer on this point, and have now added it to the discussion.

Reviewer 2 Report
Reponses to the reviewers:
I agree that the manuscript was generally improved
I still find the argument regarding the filters relatively weak (R2, p. 3) - "are similar to other studies in our lab and others (e.g. Loui et al, 2009; Przysinda et al, 2017)". Even though these studies have been through a peer-review process this doesn't necessarily mean that the filter settings are appropriate. A minimum requirement would be to discuss this as a limitation.
I also tend to disagree on the argument in the response letter "The difference between amplitude of the N1 across conditions is not uncommon among studies in the same field that compare across different subjects and across different listening conditions; see for example Maidof and Koelsch, 2011 (Figure 2)."
The mentioned Fig. 2 compares music without or with speech which accounts for the observed N1 difference. The current study always presents music and language concurrently, the argument therefore misses the point.
---
Small mistakes:
p. 1: „attention modulates the similarity between neural resources for music and language“
the intention is understandable but it is very weird English, maybe something like „attention modulates the connection between neural resources processing music and language“?
p. 2 – split sentence: „Musical syntax processing has been localized to the inferior frontal gyrus (IFG) in magnetoencephalographic (MEG) and fMRI studies (Bianco et al., 2016a; Cheung,
Meyer, Friederici, & Koelsch, 2018; Maess, Koelsch, Gunter, & Friederici, 2001; Tillmann et al., 2006). Additional results from ERAN of lesioned patients (Sammler, Koelsch, & Friederici, 2011) and in children with Specific Language Impairment (Jentschke, Koelsch, Sallat, & Friederici, 2008) have also provided causal evidence for the reliance of musical syntax on classic language-related areas such as the inferior frontal gyrus.“
Furthermore, I recommend to drop „causal“ before evidence and to add „processing“ between musical syntax and on
p. 3 – Full stop missing in l. 104
p. 3 – This can be changed anymore but using an intelligence / intellectual impairment test from the 1940 is maybe not a most appropriate way to a valid measurement. It maybe should be discussed as a limitation.
p. 4 – that the SDs (for all measures but age) are so different in the two groups is another limitation, also that the proportion of musicians and native speakers in the experiment was so different among groups
p. 7 – Table 2, ensure that eta shows, also applies to Table 3
Author Response
We are pleased that the reviewer finds the manuscript to be generally improved. We now explain in the Methods section that the filtering settings were chosen based on previous work that looked at target ERPs similar to the current study, and we acknowledge that the filter settings can introduce artifacts in EEG data, citing Widmann et al (2015).
We acknowledge that the difference between N1 amplitudes in the Attend-Music and Attend-Language conditions is difficult to interpret. Although the stimulus materials are the same except for the manipulation of attention to language vs music, the fact that the Attend-Language and Attend-Music conditions were conducted between-subjects makes it difficult to interpret whether the N1 differences arise from task-related effects or from subject effects. We now acknowledge the need for matching the EEG baselines between groups in the limitations section.
p. 1: We have changed that sentence to “attention modulates the shared processing of music and language”, to be more consistent with the original claims of SSIRH.
p. 2 : Done.
p. 3 : Fixed.
p. 3 : We agree with the reviewer that it could be a limitation that a screening tool for intellectual impairment that was developed in 1940 may no longer have the same sensitivity today. This need to continually validate and match our samples with sensitive baseline measures is now added in the limitations section.

This manuscript is a resubmission of an earlier submission. The following is a list of the peer review reports and author responses from that submission.
Round 1
Reviewer 1 Report
Lee and colleagues investigate the role of attention for the interaction of syntactic processes in music and language. They adopt the paradigm by Slevc et al. (2009) that crosses visually presented garden-path sentences with key violations in auditory chord sequences. In two separate groups, participants either monitored the correctness of the musical or language channel. The authors observed a pre-attentive ERAN followed by an attention-dependent late positivity to music-syntactic violations, whereas linguistic garden-path sentences evoked late effects only, and only in the attend-language group. The authors conclude on early automatic and late controlled processes that differ between music and language. The study directly addresses the Shared Syntactic Integration Resource Hypothesis (SSIRH) and the recently raised idea that what is shared between domains is not necessarily syntactic but domain-general processes (Slevc & Okada, 2015, cited by the authors). This is an interesting question, however, I do have several concerns that should be addressed in a major revision. I’ll specify those points below. [1] Crossing music and language. The original study by Slevc et al. (2009) investigated the interaction between syntactic violations in music and language. This point is emphasized particularly strongly in the introduction (e.g., lines 100-102), raising the expectations in readers to see a similar analysis in the present study (also similar to Koelsch et al., 2005, cited by the authors). However, it looks like no interaction was tested – how did the ERAN, P600 and N400 differ depending on the syntactic structure of the respective other channel? This needs some streamlining to balance introduction and discussion (where the interaction is no longer relevant). Please also specify the paradigm, for example, which conditions were presented and with how many trials. The text reads as if there were 48 trials in total – which would be 12(?) trials for each of the 4(?) conditions (No violation, Music violation, Language violation, Double violation). [2] Hypotheses. The authors explain which ERP components are typically evoked during processing of musical and linguistic syntax, but do not refer specifically to effects previously observed with the paradigms used here. For example, it is not clear why the authors expect to find an ELAN in garden path sentences in the first place (I am not aware of such effects in garden path sentences). The study by Osterhout et al. (1994) may be particularly relevant here, but also later publications. Likewise, it would be helpful to know which ERP components have been reported for the present musical paradigm so far. The study by Patel et al. (1998) may be relevant here, for example. Altogether, it may be good to tailor the description of ERP components more to the present paradigms. [3] Discussion. Given that no interaction was tested, I find it problematic to conclude on “separate neural resources for music and language” (abstract). Likewise, the conclusion that “Attention affects musical and linguistic syntax processing differently at late time windows” (line 257) is somewhat misleading. It is not necessarily attention that made the difference, but finding different ERP components in a language and music experiment may simply be due to the fact that the two paradigms tap into distinct sub-processes. Garden-path sentences not necessarily evoke early effects (I am not aware of an ELAN or LAN effect in garden-path studies) but have been described to trigger late re-analysis/repair processes reflected in a P600 (that may interact with late positive effects in music). Likewise, music-syntactic violations evoke either a P300 or an N500, depending on the task and musical expertise of the listeners (compare Koelsch et al., 2005, J Cogn Neurosci; Steinbeis & Koelsch, 2008, Cereb Cortex; Steinbeis & Koelsch, 2006, J Cogn Neurosci). The authors may wish to specify which processes they think have been addressed. [4] Referencing is partly imprecise. For example, Cheung et al. (line 68) is not an MEG experiment, Sun et al. (2018) did not test lesion patients (but congenital amusics) (line 70). Jentschke et al. (2008) did not test lesion patients either, but children with specific language impairment (line 78). Please add some fMRI references in line 71 showing IFG involvement in music-syntactic processing (e.g., Tillmann et al., 2006; Koelsch et al., 2005; Bianco et al., 2016). Sammler, Koelsch, & Friederici is cited in two versions (e.g., lines 70, 79) – the 2011 version should be kept. I was surprised to not see a reference to Patel et al. (1998, J Cogn Neurosci) who showed a P600 (or P3) in both music and language. Minor points: - line 33: correct “Shared Syntactic Integration Resource Hypothesis” – no s in Resource. - line 157: please briefly mention what “Shipley” and “MBEA” measure. What does “pitch perception” mean? - line 169: What is “experiment 2”? - line 221/227: spell out “topos” as “topographies” - Table 2: Please change table 2 in the following way – delete columns SS and MS and rows Error (Violation), Intercept and Error. Please add a column with effect sizes, e.g. partial eta squared. - Table 2: the first p-value in the table (pAuthor Response
R1
[1] Crossing music and language. The original study by Slevc et al. (2009) investigated the interaction between syntactic violations in music and language. This point is emphasized particularly strongly in the introduction (e.g., lines 100-102), raising the expectations in readers to see a similar analysis in the present study (also similar to Koelsch et al., 2005, cited by the authors). However, it looks like no interaction was tested – how did the ERAN, P600 and N400 differ depending on the syntactic structure of the respective other channel? This needs some streamlining to balance introduction and discussion (where the interaction is no longer relevant). Please also specify the paradigm, for example, which conditions were presented and with how many trials. The text reads as if there were 48 trials in total – which would be 12(?) trials for each of the 4(?) conditions (No violation, Music violation, Language violation, Double violation).
We thank the reviewer for this very important point. We did not have a double-violation condition because we were mainly interested in comparing neural responses between the two processes, rather than the interaction between the two processes. The main question motivating our study is “does the brain process music violation and language violation similarly?” rather than “does the brain process simultaneous music violation and language violation more than either violation alone?” We now reword the introduction to clarify this. We also include a more extensive introduction of our attention hypothesis upfront: Although we did not have a double-violation condition, we do test an interaction between attention and violation in each modality, because our hypothesis is that attention has an effect on the processing of musical syntax and linguistic syntax.
We now also provide more information about the paradigm: there were 144 trials in the experiment: 48 no violation, 48 music violation, and 48 language violation. Participants were randomly assigned to Attend-Music and Attend-Language groups.
[2] Hypotheses. The authors explain which ERP components are typically evoked during processing of musical and linguistic syntax, but do not refer specifically to effects previously observed with the paradigms used here. For example, it is not clear why the authors expect to find an ELAN in garden path sentences in the first place (I am not aware of such effects in garden path sentences). The study by Osterhout et al. (1994) may be particularly relevant here, but also later publications. Likewise, it would be helpful to know which ERP components have been reported for the present musical paradigm so far. The study by Patel et al. (1998) may be relevant here, for example. Altogether, it may be good to tailor the description of ERP components more to the present paradigms.
We thank the reviewer for pointing this out. We agree with the reviewer that P600, rather than ELAN, is the correct hypothesized time window given our garden path sentence stimuli. We now tailor the description of ERP components more to fit with these hypotheses in our introduction. Since we did not observe a significant P600 in response to the garden path sentences during attend-language condition (unlike Patel et al, 1998, and Osterhout et al, 1994), but we did observe a P600-like effect in response to the music violation during the attend-music condition (replicating Patel et al, 1998), we now also discuss these results in light of the literature on N400 and P600 in the discussion.
[3] Discussion. Given that no interaction was tested, I find it problematic to conclude on “separate neural resources for music and language” (abstract). Likewise, the conclusion that “Attention affects musical and linguistic syntax processing differently at late time windows” (line 257) is somewhat misleading. It is not necessarily attention that made the difference, but finding different ERP components in a language and music experiment may simply be due to the fact that the two paradigms tap into distinct sub-processes. Garden-path sentences not necessarily evoke early effects (I am not aware of an ELAN or LAN effect in garden-path studies) but have been described to trigger late re-analysis/repair processes reflected in a P600 (that may interact with late positive effects in music). Likewise, music-syntactic violations evoke either a P300 or an N500, depending on the task and musical expertise of the listeners (compare Koelsch et al., 2005, J Cogn Neurosci; Steinbeis & Koelsch, 2008, Cereb Cortex; Steinbeis & Koelsch, 2006, J Cogn Neurosci). The authors may wish to specify which processes they think have been addressed.
We agree with the reviewer and have reshaped our discussion section. Rather than concluding that our findings of different ERP components reflect separate neural resources for music and language per se, we now add the interpretation that the garden-path sentences and music-syntactic violations used in our present study tapped into distinct sub-processes of syntax processing. We have now reworded the abstract and added this point to the discussion.
[4] Referencing is partly imprecise.
We thank the reviewer for pointing out these inaccuracies; they are now corrected and the appropriate references are now added.
Minor points: - line 33: correct “Shared Syntactic Integration Resource Hypothesis” – no s in Resource.
Now fixed.
line 157: please briefly mention what “Shipley” and “MBEA” measure. What does “pitch perception” mean?
Thanks for helping us clarify these. The two groups of subjects did not differ in terms of general intellectual ability, as measured by the Shipley Institute of Living scale for measuring intellectual impairment and deterioration (Shipley, 1940). Nor did the two groups differ in low-level pitch discrimination abilities as assessed by a pitch-discrimination task of two-up-one-down staircase procedure around the center frequency of 500 Hz (Loui, Guenther, Mathys, & Schlaug, 2008)). They also did not differ in years of musical training (years of musical training was not different between the two groups, X2= 0.0215, p = .88). These are now clarified in the manuscript.
line 169: What is “experiment 2”? –
By “experiment 2” we meant the Attend-Music condition; this is now corrected for consistency.
line 221/227: spell out “topos” as “topographies” –
Done.
Table 2: Please change table 2 in the following way – delete columns SS and MS and rows Error (Violation), Intercept and Error. Please add a column with effect sizes, e.g. partial eta squared. –
Done.

Reviewer 2 Report
The manuscript focussed on exploring the effect of attention on the processing of syntax violations in music and language. If I understood the methods correctly, it is mainly a replication of the Slevc et al. (2009)-study complementing the behavioural results with neurophysiological measurements.
Generally, I have to say I found difficult to judge whether the study really explored what it was supposed to explore given the unclear description in the methods section and a (possibly major) methodological flaw: It appears as if language and music were presented concurrently. If that was the case then I don't understand how the four conditions arise and what they represent (music congruent, music incongruent, language congruent, and language incongruent). I also come up with four conditions but don't know how they match with the four conditions in the study ([1] music expected - language expected; [2] music expected - language unexpected; [3] music unexpected - language expected; [4] music unexpected - language unexpected). One would then end up with four different results: [1] neither ERAN nor ELAN; [2] ELAN; [3] ERAN; [4] interaction of ERAN and ELAN (i.e., a 2 x 2 design with musical syntax violation as one factor and linguistic syntax violation as the other). This 2 x 2 design would then have to be further explored using the between subject factor attention (to music vs. to language). I don't really know or understand what the current analysis is evaluating.
Some further comments:
- The article should mention: Maidhof, C., & Koelsch, S. (2011). Effects of selective attention on syntax processing in music and language. Journal of cognitive neuroscience, 23(9), 2252–2267. https://doi.org/10.1162/jocn.2010.21542 who explored the same research question using a dichotic listening paradigm.
- Currently, the sample includes L2-speakers of English. This influences the automaticity of the ELAN and is hence confounded with attention. In my view it would be better to remove these participants. The sample also is unbalanced when it comes to gender and musical training (influencing the ERAN-amplitude size; eg. Koelsch et al., 2002; Jentschke & Koelsch, 2009).
- Even though linguists traditionally separate syntax and semantics, they might be less independent than this separation suggests (see, e.g., articles from Hagoort). This also might partially account for the results of Perruchet & Poulin and should be discussed.
- When using out-of-key chords the music-syntactic violation is confounded with a sensory violation.
- Recording and filter conditions seem a bit odd to me. Why sampling with 1000 Hz to later low-pass filter the data at 30 Hz. Furthermore, the cut-off for high-pass might affect the ERAN amplitude (cf.,Weber, C., Hahne, A., Friedrich, M., & Friederici, A. D. (2004). Discrimination of word stress in early infant perception: electrophysiological evidence. Cognitive Brain Research, 18(2), 149–161. https://doi.org/10.1016/j.cogbrainres.2003.10.001; Widmann, A., Schröger, E., & Maess, B. (2014). Digital filter design for electrophysiological data - a practical approach. Journal of Neuroscience Methods, 250, 34–46. https://doi.org/10.1016/j.jneumeth.2014.08.002).
- When looking at Figure 2, the data generally seem quite noisy and most importantly the components reflecting early auditory processing (P1/N1 complex) differ substantially (e.g., is the N1-amplitude ~1.5 µV in panel A and ~0.5 µV in panel B). This makes me doubt whether the later ERP components are reliable.
I did not bother to read the discussion given that I had doubts whether design and statistical analysis were sound.
Author Response
R2
Generally, I have to say I found difficult to judge whether the study really explored what it was supposed to explore given the unclear description in the methods section and a (possibly major) methodological flaw: It appears as if language and music were presented concurrently.
We thank the reviewer for this comment. We do not believe it was a methodological flaw that language and music were presented concurrently, because the syntactic violations of music and language were not presented concurrently. In all our trials, both modalities were either fully congruent, or only one of the modalities was incongruent. As in the response to R1, there was no dual-violation condition, because the goal of this study is to directly compare music and language expectations rather than to compare each violation against a simultaneous music and language violation. Thus, there were three conditions: no violation, music violation, and language violation.
As the other reviewer noted, ELAN is not expected with garden path sentences; instead, the previous literature has shown a P600 (late positive complex). We now note that our hypotheses pertain to two windows (early and late). In the early tine window we expect the ERAN (during music violations), and in the later time window we expect N400 (during language violations), or LPC/P600/P300 (during both language and music violations). In this resubmission we spell out these hypothesized ERP effects.
- The article should mention: Maidhof, C., & Koelsch, S. (2011). Effects of selective attention on syntax processing in music and language. Journal of cognitive neuroscience, 23(9), 2252–2267. https://doi.org/10.1162/jocn.2010.21542 who explored the same research question using a dichotic listening paradigm.
We thank the reviewer for pointing out this article; this is now reviewed in the Introduction section.
- Currently, the sample includes L2-speakers of English. This influences the automaticity of the ELAN and is hence confounded with attention. In my view it would be better to remove these participants. The sample also is unbalanced when it comes to gender and musical training (influencing the ERAN-amplitude size; eg. Koelsch et al., 2002; Jentschke & Koelsch, 2009).
We thank the reviewer for pointing out this potential confound. We now reanalyze the data with first-language English speakers only, and show that removing the L2 speakers did not significantly alter our results. This is now added to the results section (see Table 3).
- Even though linguists traditionally separate syntax and semantics, they might be less independent than this separation suggests (see, e.g., articles from Hagoort). This also might partially account for the results of Perruchet & Poulin and should be discussed.
We agree with the reviewer that syntax and semantics may not be completely independent. We now discuss this in the Discussion section.
- When using out-of-key chords the music-syntactic violation is confounded with a sensory violation.
We appreciate the reviewer’s point that music-syntax violations often co-occur with sensory violations. In this study, however, the out-of-key chords are carefully constructed to avoid sensory dissonance and roughness (see Slevc et al, 2009). Thus the effects cannot be explained by sensory violations. Furthermore, Koelsch et al (2007) had shown that ERAN is elicited even when irregular chords are not detectable based on sensory violations, which supports the role of ERAN in music-syntax violations. Given our stimuli as well as previous evidence, we believe that the currently observed ERAN reflects music-syntax violations rather than sensory violations. This point is now added to this Discussion section.
- Recording and filter conditions seem a bit odd to me. Why sampling with 1000 Hz to later low-pass filter the data at 30 Hz. Furthermore, the cut-off for high-pass might affect the ERAN amplitude (cf.,Weber, C., Hahne, A., Friedrich, M., & Friederici, A. D. (2004). Discrimination of word stress in early infant perception: electrophysiological evidence. Cognitive Brain Research, 18(2), 149–161. https://doi.org/10.1016/j.cogbrainres.2003.10.001; Widmann, A., Schröger, E., & Maess, B. (2014). Digital filter design for electrophysiological data - a practical approach. Journal of Neuroscience Methods, 250, 34–46. https://doi.org/10.1016/j.jneumeth.2014.08.002).
The present recording and filter conditions are similar to other studies in our lab and others (e.g. Loui et al, 2009; Przysinda et al, 2017). Sampling rate and filters are independent steps, after having acquired data at a sampling rate above the Nyquist rate. It is customary to record with high sampling rate and then low-pass filter to remove high-frequency noise, which can result from electrical noise as well as muscle artifacts (Luck, 2014). By starting with a high sampling rate of 1000 Hz, we start with the highest quality data, then remove the frequency bands that are A) out of the range of our frequencies of interest, and B) susceptible to contamination with sources of noise unrelated to the study. While the reviewer is correct that high-pass and low-pass filtering can affect the amplitudes of specific ERP potentials, we note that these steps were the same across all our experiment conditions and all task manipulations; thus the differences observed in the ERPs across conditions and across task manipulations cannot be explained by our filtering choices.
- When looking at Figure 2, the data generally seem quite noisy and most importantly the components reflecting early auditory processing (P1/N1 complex) differ substantially (e.g., is the N1-amplitude ~1.5 µV in panel A and ~0.5 µV in panel B). This makes me doubt whether the later ERP components are reliable.
The data are not noisy in that the pre-stimulus noise is much smaller than the ERPs that reflect sensory processes (e.g. P1/N1 complex). The difference between amplitude of the N1 across conditions is not uncommon among studies in the same field that compare across different subjects and across different listening conditions; see for example Maidof and Koelsch, 2011 (Figure 2).
